# Divergent evolution of slip banding in CrCoNi alloys

Bijun Xie[1,3], Hangman Chen[1,3], Pengfei Wang [1], Cheng Zhang [2], Bin Xing[2], Mingjie Xu[2], Xin Wang[2], Lorenzo Valdevit [1,2], Julian Rimoli[1], Xiaoqing Pan [2] & Penghui Cao [1,2] ✉

Metallic materials under high stress often exhibit deformation localization, manifesting as slip banding. Over seven decades ago, Frank and Read introduced the well-known model of dislocation multiplication at a source, explaining slip band formation. Here, we reveal two distinct types of slip bands (confined and extended) in compressed CrCoNi alloys through multi-scale testing and modeling from microscopic to atomic scales. The confined slip band, characterized by a thin glide zone, arises from the conventional process of repetitive full dislocation emissions at Frank–Read source. Contrary to the classical model, the extended band stems from slip-induced deactivation of dislocation sources, followed by consequent generation of new sources on adjacent planes, leading to rapid band thickening. Our findings provide insights into atomic-scale collective dislocation motion and microscopic deformation instability in advanced structural materials.

Deformation banding, where strain concentrates in local zones, is ubiquitous in many human-made and natural systems, ranging from crystalline solids[1], metallic glasses[2], and granular media[3] to geological faults[4] under compressive stress. It occurs when the external stress surpasses the elastic limit, acting as a mechanism that enables the system to mitigate stressing condition and dissipate the stored strain energy[1,5–8]. First shown by Ewing in 1900[5], the fine structure of shear bands in plastically deformed crystals was examined by Heidenreich and Shockley[7] and then by Brown[8] using electron microscope in the 1940s, and stimulated the theoretical studies of the origin of why deformation would concentrate in a band region[9,10].

These bands in plastically deformed aluminum consist of elementary slip lines separated by around ten nanometers[7,8]. Notable surface steps of several hundred nanometers appear at the intersection of individual slip lines and free surface, suggesting thousands of dislocations (line defects in crystals[11,12]) have passed through the slip plane. Intrigued by these sharp surface markings, Frank and Read[10] envisioned a dislocation source inside the material, at which dislocations can rapidly multiply and produce a vast number of dislocations in a fraction of a second[13], interpreting the band formation. When the generated dislocations become piled up against some obstacle, the back stress from these stacked dislocations opposes the resolved shear stress (from applied stress) acting on the source, pausing its loop generation[9]. As the applied stress increases or when thermally activated cross-slip[14] occurs, the ceased source restarts loop generation, therefore giving rise to the dynamic dislocation creation and slip avalanche associated with slip band evolution.

With the groundbreaking test methodology for micropillar prepared using focused ion beam revealed in 2004[15], the intermittency and size[16] of dislocation avalanche can be directly determined by measuring the slip event on the serrated plastic flow of stress-strain curve[17]. Since then, innumerable experiments at micro-and nano-scales have been taken to probe the yielding onset[18], dislocation avalanche[19,20], and shear localization[21,22]. Over 70 years after the Frank-Read, however, the atomic-scale dislocation dynamics and their correlation with slip banding (how coordinated dislocation activities drive and evolve into shear bands) remain an open question. The overlook of slip banding may be due to its natural occurrence under mechanical straining or the widespread acceptance of Frank-Read model. This is pressing because even the fine band structure frequently observed in aluminum has long been noticed that it rarely appears in many metals and alloys[23,24].

[1]Department of Mechanical and Aerospace Engineering, University of California, Irvine, CA, USA. [2]Department of Material Science and Engineering, University of California, Irvine, CA, USA. [3]These authors contributed equally: Bijun Xie, Hangman Chen. ✉e-mail: caoph@uci.edu

In this study, we perform in-situ mechanical compression of single-crystal CrCoNi micropillars and unveil two distinct slip bands, here referred to as confined slip band (C-SB) and extended slip band (E-SB). Analyzing band structures from microscopic down to atomic scales using scanning transmission electron microscopy (STEM) and four-dimensional (4D) STEM characterizations, we find the C-SB, manifesting as a thin glide zone, is scarce in defects, yet the E-SB embodies a high density of planar defects. In accordance with the Frank-Read theory, the C-SB originates from dislocation multiplication at active sources. Surprisingly, we find the E-SB results from dislocation slip-induced deactivation of the Frank-Read source and the subsequent dynamic generation of new sources. Our findings elucidate a critical phenomenon in strained materials that slip band initiating from a common dislocation source evolves divergently, governed by the underlying dislocation slip multiplicity.

## Results

### Confined slip band

To activate full dislocation slip, we prepare [110] oriented single-crystal micropillars for mechanical testing (Supplementary Fig. 1). Compressing along this orientation, two of the four slip planes, specifically (111) and ($\bar{1}11$), (i.e., ABC and ABD in Thompson tetrahedron notation in Fig. 1a) can be activated due to their largest Schmid factor. In this orientation, dislocations presumably operate in pairs (i.e., full dislocation mechanism that includes leading and trailing partial dislocations), because the Schmid factor of trailing partials (-0.47) can be twice as high as that of leading partials (-0.24). Figure 1b shows the plastically deformed sample, exhibiting stepped surface and deformation strips. To further analyze these features, we section the micropillar to examine the surface step and characteristics of slip bands. The bright-field STEM (BF-STEM) image in Fig. 1c reveals multiple sharp surface steps, suggesting a highly concentrated slip in these ultra-thin bands. The steps, ranging in size from 15 to 135 nm, indicate that approximately 60 to 500 dislocations have been released on these stepped surfaces. Different from the fine band structure in aluminum[7,8], the spacing of these slip lines appears to have a large variation and an irregular distribution (Supplementary Fig. 3). This irregularity in location can be related to the predominant planar slip nature of CrCoNi and the random location of the dislocation source that the material contains.

We probe the bulk interior along the stepped surface to delineate the defect structures that underpin the C-SBs. Figure 1d shows the structural features, as characterized by both bright field-TEM and high-angle annular dark-field STEM (HAADF-STEM), covering length scales from submicron to atomic levels. Along the slip band, which corresponds to the primary slip plane of ABC in the Thompson tetrahedron, defects are remarkably sparse. This indicates that the band formation is driven by full dislocations, which glide without leaving defects behind. The dislocations presumably nucleate from the bulk interior at Frank-Read sources, where the nucleation stress is substantially smaller than that of the surface step (Supplementary Fig. 6 and Supplementary Note 4). On the secondary slip plane ABD that intersects with ABC, a dense array of stacking faults (SFs) is evident (Fig. 1d and Supplementary Fig. 4), implying an increased separation between leading and trailing partial dislocations. This observation can be interpreted by slip-induced pillar rotation[25], which favors full dislocation activity on the primary slip plane but partial dislocation on the secondary slip plane (Fig. 1a). Notably, the SFs exhibit a kinked structure across the slip band. These kinks result from dislocation slip along the ABC, which displaces the exiting SFs by one Burgers vector after each loop passage (Supplementary Fig. 5).

### Extended slip band

To examine how partial dislocation slip impacts shear band formation, we fabricate micropillars with [100] orientations aligned with the compression direction. For all four non-parallel slip planes containing the slip systems, the leading partial has the Schmid factor of 0.47, nearly doubling that of its trailing counterpart (Fig. 2a). Plastic deformation is presumably mediated by partial dislocations, especially given the low SF energy. Figure 2b shows the plastically deformed structure with primary slip occurring on the ABD plane. This selective slip activation is associated with micropillar rotation during deformation, which favors one slip plane while simultaneously disfavoring the other three (Fig. 2a). In contrast to the slip-concentrated bands (Fig. 1), the strained system exhibits widened deformation strips, appearing as ribbon-like band structure (Fig. 2b). In the plastically deformed zone, extended slip bands emerge with a large thickness from 150 to 320 nm. Upon examining the surface (Supplementary Fig. 7), fewer steps are visible only at the junctions between bands. The deformation surface of these extended slip bands is relatively smooth yet tilted, indicating a more uniform glide of dislocations within these bands.

Figure 2d highlights three representative deformation regimes. Regime i represents the boundary of the E-SB, i.e., the interfacial region between non-slipped matrix and the slip band. Atomic-scale imaging elucidates that within this deformation structure, a pronounced deformation twin is manifested, bounded by twin boundaries (Fig. 2e). Utilizing 4D-STEM, the twin thickness has been characterized as more than 100 nm, determined by its crystallographic orientation relative to the adjacent matrix (Supplementary Fig. 8). Within the bulk interior of the deformation band, regime ii is characterized by diverse deformation microstructure (Fig. 2f), which encompasses an array of twinned lamellae, interspaced by relatively thin hexagonal close-packed (hcp) lamellae and SFs. Strikingly different from the C-SB, which retains a nearly pristine structure, the E-SB features densely packed planar defects interspersed in the entire deformation band, indicating the result of partial dislocation slip. In regime iii of the band-band interface (Fig. 2g), there is a high concentration of planar defects, appearing as twin and hcp lamellae, and SF layers. The abundance of these planar defects is likely associated with the two shear bands, where emitted partial dislocations drift to and accumulate at this boundary. It is important to note that these planar defects, including deformation twins, hcp lamellae, and SFs, result from dislocation slip, the fundamental and elementary deformation mechanism governing the microstructure evolution. The dislocations are found to nucleate from bulk rather than the surface, as the nucleation stress in the bulk source is significantly lower than at the surface, even with steps (Supplementary Fig. 9 and Supplementary Note 4).

### Plastic strain progression and microstructure transformation

Utilizing large-scale atomistic simulations, we elucidate spatiotemporal development of local plastic strain and microstructural transformation within a crystalline pillar under compression. We introduce inherent vacancy defects in the bulk of materials, which act as dislocation nucleation sites (Frank-Read sources, see Supplementary Figs. 10, 11). This setting effectively captures dislocation multiplication within the bulk pillar while mitigating the impact of free surface effects on plasticity that is prevalent in nanocrystals. Figure 3a shows the compressed [110] oriented pillar, where the local strain map and microstructure are presented. At an early stage of plastic deformation (5% strain), a few strain-localized lines emerge on the ABC and ABD planes, indicating the passage of dislocations. Microstructure analysis of these areas reveals a retained fcc structure, confirming an activation of a full dislocation mechanism. As the strain increases to 25%, a prominent strain-concentrated band (i.e., C-SB) develops along the primary slip plane, accumulating substantial plastic strain, as also evidenced by the sharp surface step. Intriguingly, the [100] oriented pillar reveals a markedly different strain map and microstructure pattern (Fig. 3b). At the same strain level of 25%, a wider and thicker expansive band (i.e., E-SB) emerges. The band, terminated by TBs at

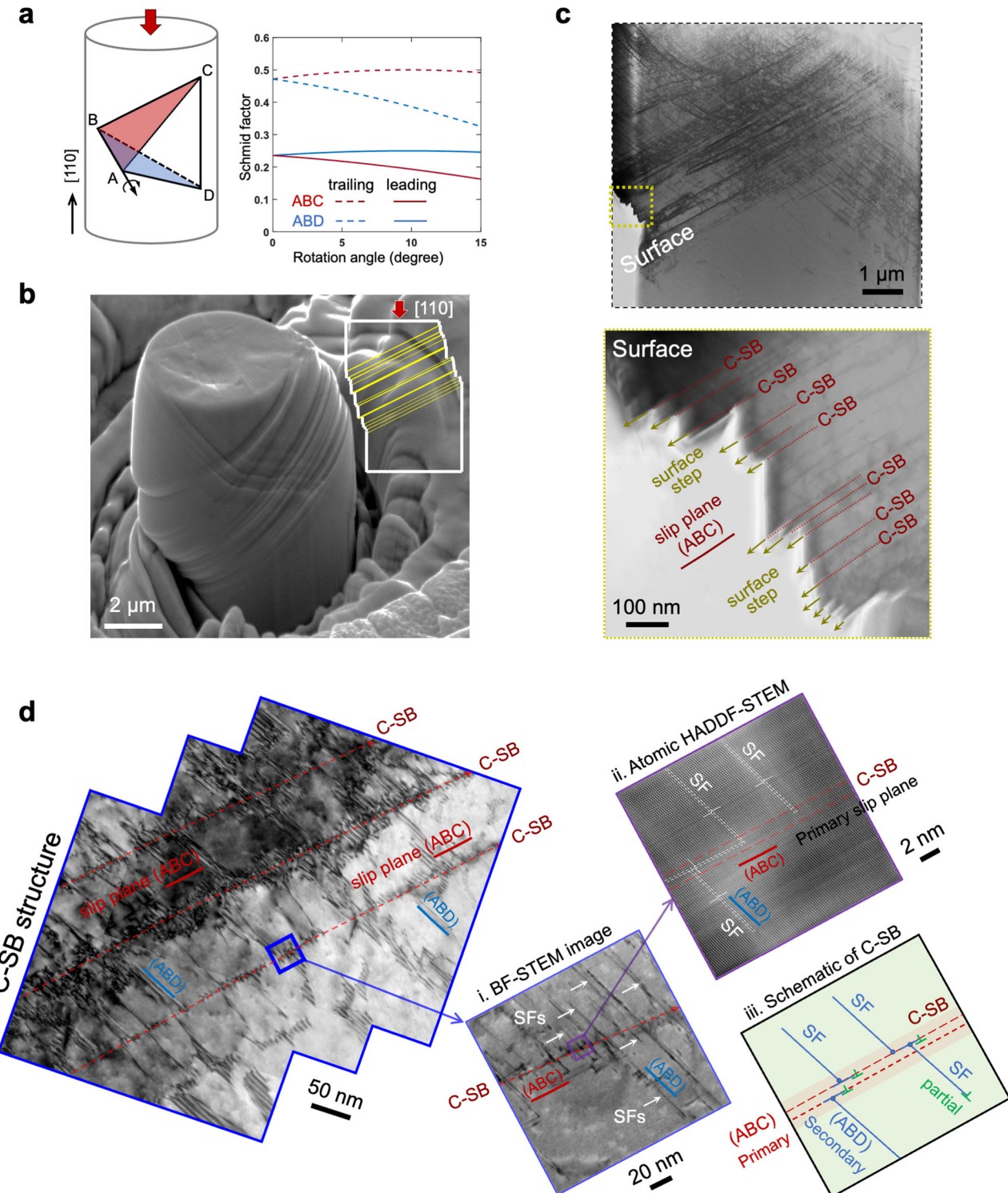

**Fig. 1 | Confined slip band (C-SB) and its microscopic and atomic structures.** **a** Schmid factors for the slip systems of partial dislocations gliding on the ABC and ABD planes, including their changes with compression-induced rotation. Dashed and solid lines denote trailing and leading partial dislocations, respectively. **b** SEM image of [110]-oriented micropillar after plastic compression, exhibiting a stepped surface and deformation strips. **c** BF-STEM images of compressed micropillar reveal sharp surface steps, signifying highly concentrated deformation within the confined slip bands. **d** Magnified TEM images elucidate the structural features of C-SB from submicron to atomic scales. The slip band, aligned along the primary slip plane (ABC), retains a nearly pristine structure with sparse deformation defects. A dense array of stacking faults is evident along the secondary slip plane (ABD).

the boundary, hosts a high density of hcp phases and SFs (see Fig. 3d for statistics distribution of defects). The results of these simulations, congruent with our experimental characterizations of the two slip bands, clarify how plastic strain progresses and how the microstructure evolves during deformation.

Figure 3c, d compares local plastic strain and defects in the two deformed systems. In accommodating the externally applied deformation, the forming E-SB redistributes strain across an expanding zone, manifesting as a relatively small and more uniform strain profile. The C-SB, however, absorbs the strain in a more confined area,

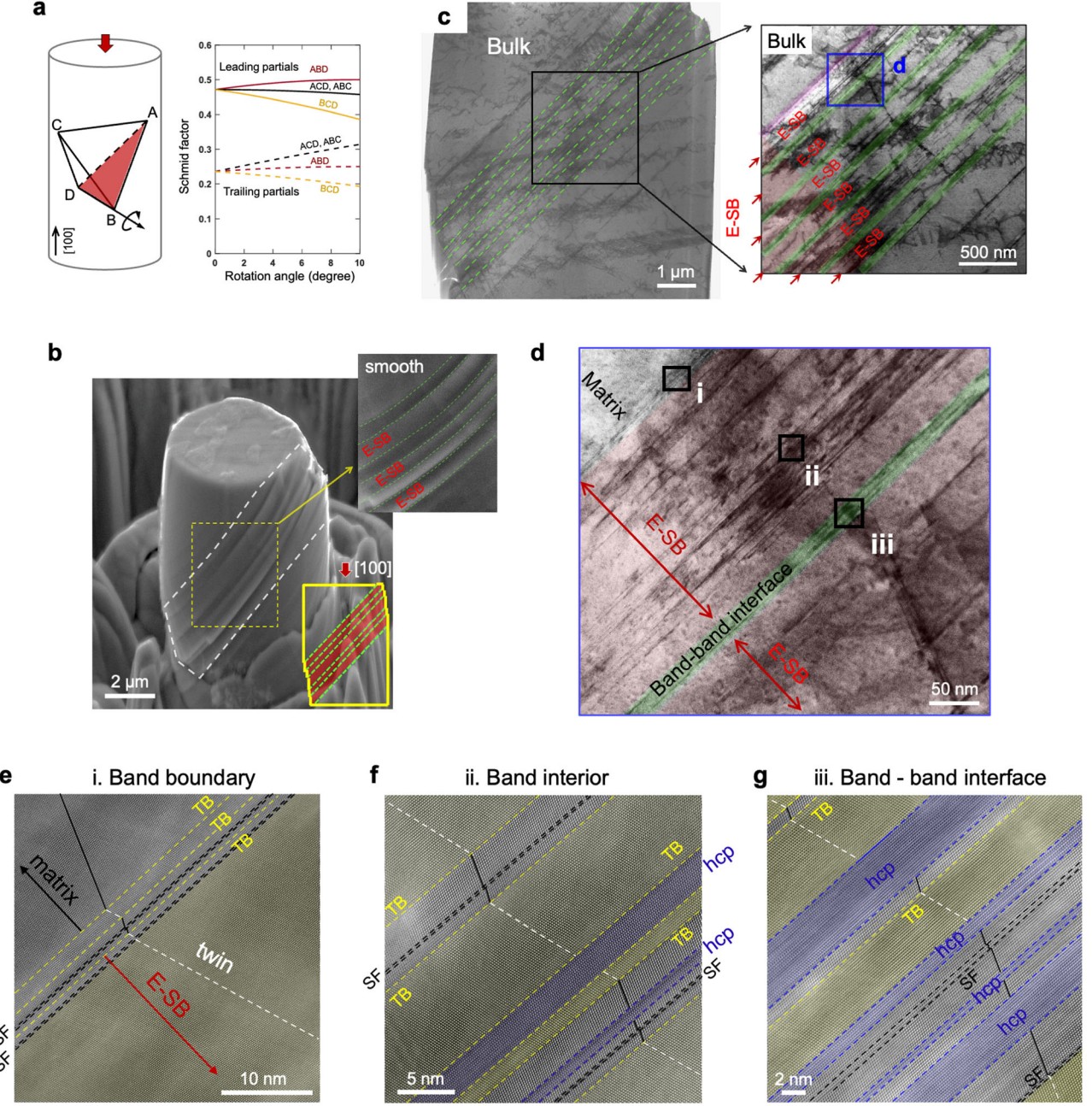

**Fig. 2 | Extended slip band (E-SB) and the microscopic and atomic structures.**
**a** Schmid factors for partial dislocations gliding on the four non-parallel planes and their variations with compression-induced rotation. Solid and dashed lines denote leading and trailing partial dislocations, respectively. **b** SEM image of [100]-oriented micropillar after plastic compression, showing widened deformation strips. **c** BF-STEM images exhibit ribbon-like extended slip bands along the ABD plane. The red color highlights these E-SBs. **d** Magnified BF-STEM image illustrates three representative regimes of the deformed pillar. Regime i represents the boundary of the band, regime ii for the band interior, and regime iii for the interface between two bands. **e–g** Atomic resolution HAADF-STEM elucidates the defects in the three regimes. The E-SB comprises a series of nano-twins, interspaced by thin hcp lamellae and stacking faults.

resulting in a large strain tail in the strain distribution (Fig. 3c). Such concentrated strain accumulation can pose a greater risk to the material integrity by potentially increasing the chances of incoherent deformation and the initiation of nano cracks, particularly in poly-crystalline materials.

## Common origin and divergent evolution of slip banding
The clear distinction between C-SB and E-SB raises an intriguing question about their respective formation pathway and the applicability of the Frank-Read model in explaining both phenomena. It is conceivable that, as the applied stress increases, the activated

dislocation sources emit a dislocation segment anchored at two pinning points. When the Schmid factor of the trailing partial dislocation surpasses that of the leading one, glide occurs as a full dislocation (pair of partial dislocations), due to the higher Peach-Koehler force acting on the trailing partial[26], as illustrated in Fig. 4a. The gradual increase in stress drives the dislocation lines to bow out and expand, eventually forming complete loops and leaving new dislocation segments at the original source (bottom panel of Fig. 4a). This loop, detached from the source, continues to expand and ultimately annihilates at the surface, contributing a surface step with a magnitude of one Burgers vector. To elucidate dislocation multiplication at the source and the resulting slip

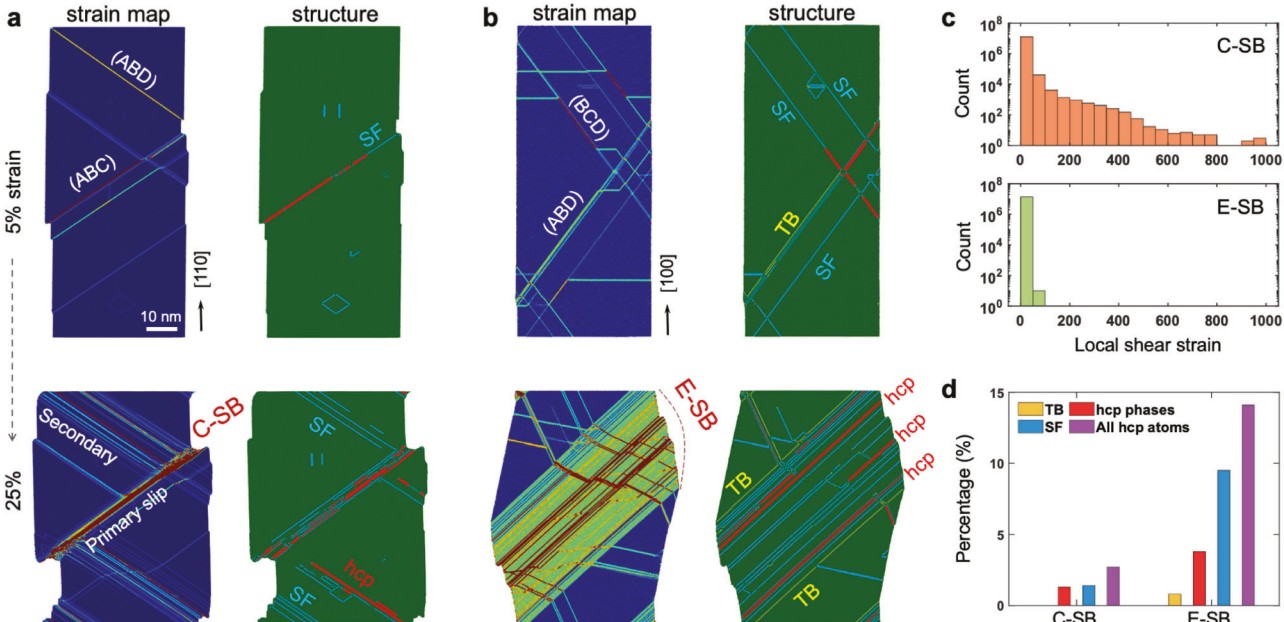

**Fig. 3 | Local plastic strain progression and microstructural evolution.**
**a**, **b** Evolution of local plastic strain and structure with increasing strain for pillars forming confined slip band and extended slip band, respectively. The structure is depicted as follows: green color represents the fcc structure, yellow denotes TB, blue indicates SF, and red represents the hcp phase. **c** Statistical distributions of local strain for pillars compressed at 25% strain. **d** The corresponding atomic fraction of TB, SF, hcp phase. The category of 'all hcp atoms' summarizes all three types of planar defects. **c**, **d** The atomic strain and structure type are analyzed for the entire deformed system, which consists of 14 million atoms.

band, we carefully prepare bulk dislocation sources in the pillar (see Methods) and conduct atomistic simulations of deformation. Figure 4b, c depicts the microstructural evolution and the progression of slip band formation with increasing strain. An activated source repetitively produces dislocations that lead to a thin slip band (i.e., C-SB) and sharp, stepped surface markings. This process of dislocation multiplication and slip band development (Fig. 4b, c) is consistent with the mechanisms proposed by Frank-Read. Due to the low SF energy in CrCoNi, cross-slipping is infrequent. Dislocations predominately navigate on the plane of their source, and band thickening occurs rarely.

The Frank-Read model, impeccably interpreting C-SB formation, encounters challenges in elucidating E-SB. We hypothesize that, if active sources consistently form at the band boundaries, continuous slip along the boundary layers can lead to band thickening. To reveal the mechanistic origin of the extended slip band and to validate the hypothesis, we perform atomistic simulations and examine the pivotal dislocation mechanisms. In the case of the Schmid factor of leading partial dislocation is greater than that of tailing partial, slip primarily occurs through the partial dislocation, as the trailing slip can be inactive due to its small Peach-Koehler force (Fig. 5a). Emitting from an activated Frank-Read source, the expansion of leading partial dislocation loop generates SF on its swept area, eventually leaving a new segment of the leading partial dislocation at the source. It is critical to note that an additional glide on the faulted plane (i.e., SF) by a leading partial would produce high energy, unstable stacking (Supplementary Fig. 13), thereby the leading partial is pinned by the formed SF. At this stage, the Frank-Read source is deactivated after it emits only one partial dislocation. Intriguingly, we find new dislocation sources are created, specifically at intersections of SFs and twin boundaries, as delineated in Fig. 5b. This source nucleation mechanism, following the dislocation reaction rule, facilitates repetitive and successive slip-twinning events on the twin boundary (Supplementary Fig. 14 and Supplementary Note 2). Figure 5c, d shows the ultra-large atomistic simulation of nanopillar compression, illustrating the dynamic processes of dislocation nucleation and slip band thickening. The results

show the critical role of dynamically relocating dislocation sources at SF-TB junctions, which, after emitting a partial dislocation, facilitate twin boundary migration and slip band growth by one atomic layer. While the simulations clearly demonstrate dislocation source generation and band thickening, the relatively small model size compared to experiments raises questions about its effectiveness in supporting the growth of twins extending to hundreds of nanometers. Moreover, if the discovered mechanism holds, we should expect to observe SF-intersected twin boundaries within the E-SB. High-resolution HAADF-STEM analysis of multiple deformation twins (Supplementary Fig. 15) reveals, remarkably, arrays of concentrated SF-TB junctions and steps distributed along the twin boundary, shown in Fig. 5e. This suggests that, in the bulk sample, a series of partial dislocations on the BCD plane glide and interact with the twin boundary (along the ABD plane), resulting in the formation of crowded twinning sources that collectively drive TB migration and advance rapid twin growth.

## Discussion

The two dislocation generation models, manifesting as distinct slip modes, can strongly impact dislocation dynamics and the stress-strain response of the system. In the confined slip band, the activated Frank-Read source keeps producing dislocations at the spatially fixed site. Until the expanding dislocation loops encounter obstacles or until enough loops have been generated and expanded along the slip plane, the back stress from dislocation repulsion acting on the source can be sufficient to temporarily pause dislocation generation. Slip, therefore, occurs in the form of intermittent avalanches of dislocations, produced by the Frank-Read source that emits loops in short intervals separated by periods of inactivity[9]. In contrast, the dislocation source in E-SB relocates to the adjacent layer after emitting a partial dislocation loop. This dynamic source migration allows for a continuous and smooth dislocation production process, which consequently influences the avalanche behavior. At the microscale, the size of the dislocation slip avalanche can be determined by the stress drop (burst) size in the plastic flow stage. An examination of the stress-strain curve serrations (Supplementary Fig. 2) reveals that the large stress drops

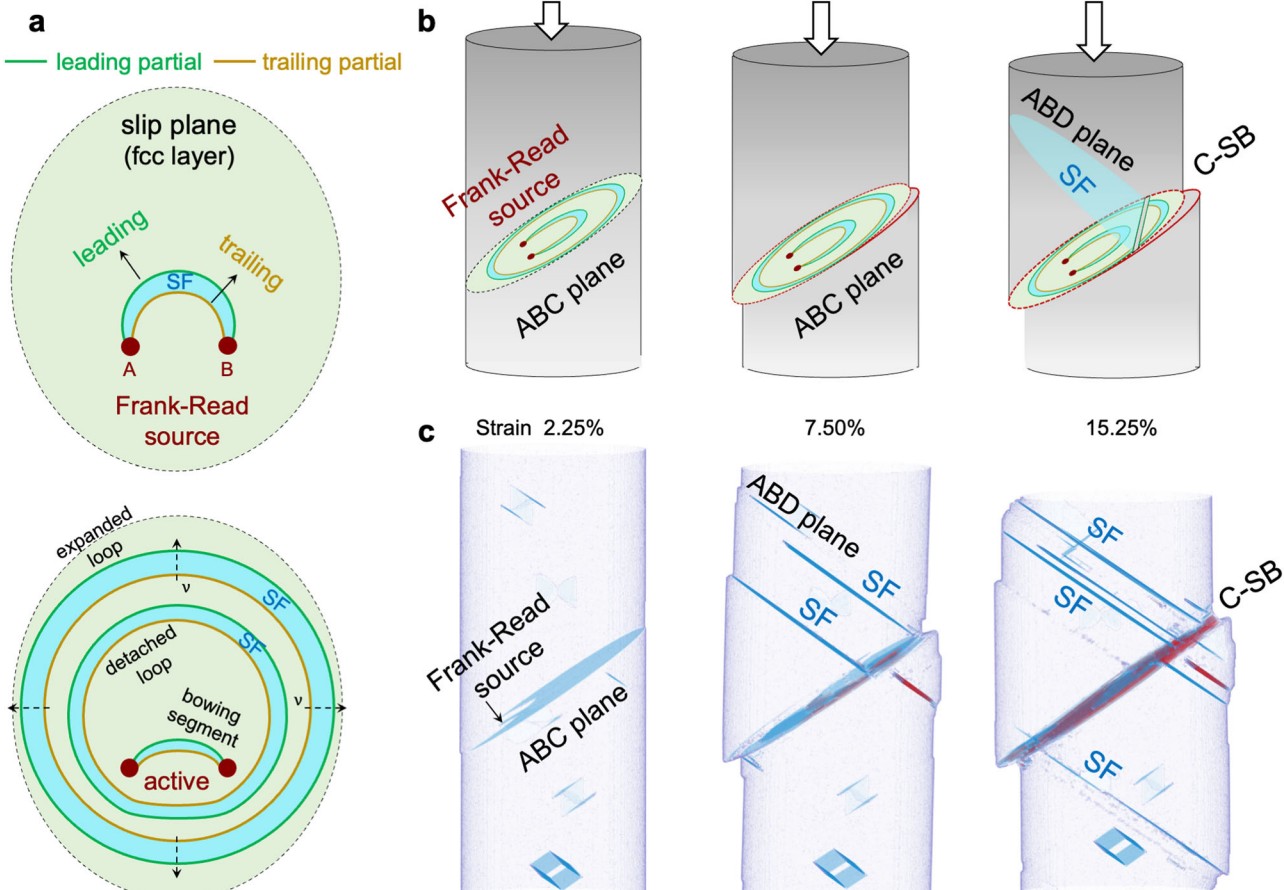

**Fig. 4 | Dislocation mechanism underpinning confined slip band formation.**
**a** Schematic illustration of dislocation multiplication at an activated Frank-Read source, where glide occurs through a pair of partial dislocations (a full dislocation mechanism), owing to the higher Schmid factor of the trailing partial dislocation compared to the leading one. **b**, **c** Atomistic modeling and schematic representation show the C-SB formation. An activated source on the ABC plane emits dislocations repetitively, forming a confined slip band accompanied by sharp surface steps. The continuous slip on the ABC plane displaces the SFs located on ABD planes, resulting in the stepped SFs at the band. In the deforming pillar (**c**), only defective microstructures are shown.

(110−480 MPa) accompanied by C-SB are substantially diminished during the plastic deformation of pillar with E-SB (Supplementary Fig. 7). This shrunk avalanches in E-SB likely stem from a more continuous dislocation generation from relocating sources, along with the extending deformation region that opens many pathways for dislocation to glide through. For the confined slip bands, however, the accumulation and jamming of a significant number of dislocations from a stationary Frank-Read source set the stage for subsequent, larger avalanches.

Another salient feature of multi-principal element alloys like CrCoNi is local chemical ordering driven by solute-solute interactions[27,28]. This local chemical order facilitated by thermal aging, whether in the form of chemical-domain structure[29,30] or ordered oxygen complexes[31] or both, raises the SF energy of the materials, presumably impacting dislocation motion and slip band evolution. To explore how aging affects slip banding and deformation localization, we prepare micropillars of the same orientation and size from aged samples for testing (Methods and Supplementary Note 3). For micropillars with the [110] orientation (full dislocation mechanism), both aged and quenched samples show a similar deformation morphology and large avalanches, indicating a minimal impact on dislocation dynamics and slip banding from thermal aging (Supplementary Fig. 17). Compared to quenched [100] micropillar (partial dislocation mechanism), interestingly, its aged counterpart exhibits more pronounced avalanches and strain localization (Supplementary Fig. 18). The different responses to thermal aging and local chemical order can be interpreted by our dislocation mechanism and slip band model. For [110] sample with the Frank-Read mechanism, full dislocations from an activated source repetitively glide through the slip plane. The presence or absence of local chemical ordering does not alter the underlying Frank-Read mechanisms and the evolution of C-SBs. However, the [100] sample, featuring an active partial dislocation mechanism, is susceptible to the presence of local chemical order. The disruption of chemical order[32] by partial dislocation slippage in the extended slip band promotes slip within this softened zone, enhancing strain localization[33]. Another possibility can be associated with the raised SF energy, which can transfer the deformation mechanism from partial dislocation to full dislocation, favoring dislocation multiplication and deformation localization. The full dislocation mechanism, when it occurs, is likely active only for a couple of slippages, after which the order disruption and lowered SF energy would change it back to a partial dislocation mechanism.

Deformation twinning is believed to take place through continuous atomic layer growth rather than occurring via a single shear−the simultaneous shuffle of all layers that would require a stress level close to theoretical strength[34]. As such, twinning is not an elementary deformation mode but rather a manifestation of a fundamental dislocation slip. The presence of hundreds of nanometer-thick deformation twins in the E-SB necessitates thousands of partial dislocations consistently and orderly gliding through the region. To enable such successive layer growth, twinning dislocation sources must appear on every slip plane. In defect-free nanowires that lack

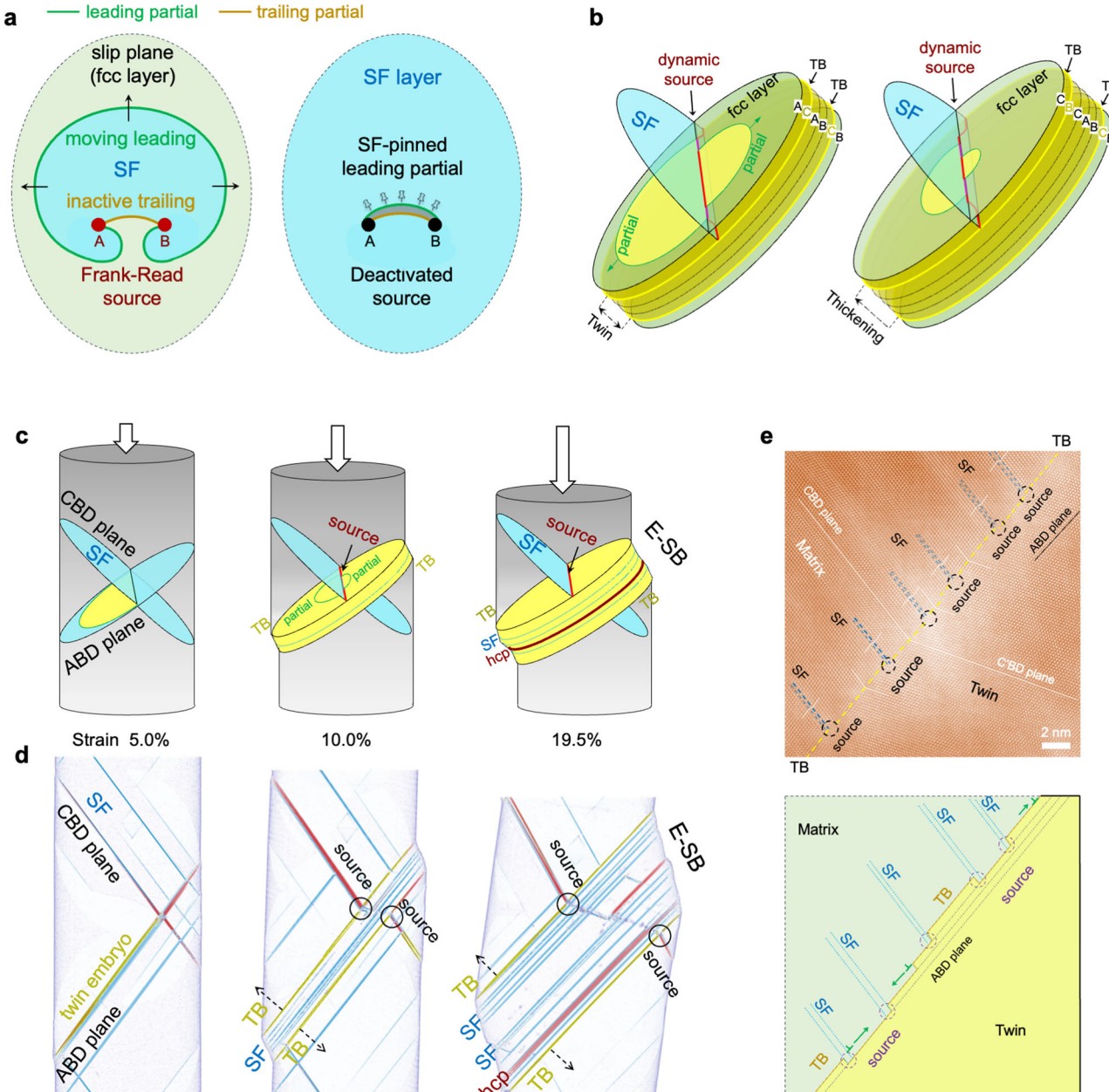

**Fig. 5 | Dislocation mechanisms driving extended slip band formation.**
**a** Schematic illustration of Frank-Read source deactivation. Expansion of a leading partial dislocation loop generates stacking fault in its traversed area, eventually leaving a new segment of the leading partial dislocation at the source. The formed SF pinning leading partial, consequently deactivating the source. **b** Schematics of dynamic dislocation sources creation. The new dislocation source, formed at intersections of stacking fault and twin boundary, moves its location with the twin boundary. **c**, **d** Atomistic modeling shows the E-SB formation. The dynamically relocating dislocation sources enable dislocation emission across successive layers, thus thickening the slip band. **d** Only the defective atoms are shown. **e** High-resolution HAADF-STEM image shows the concentrated SF-TB junctions and steps along the twin boundary. The dense twinning sources collectively drive TB migration and promote rapid twin growth at the microscale.

interior dislocation sources, the free surface acting as an active source emits partials on consecutive planes, enabling twin thickening[35]. In bulk crystals inevitably containing defects and dislocation sources, the stress required to activate dislocation at the internal sources is substantially smaller than from the surface (Supplementary Figs. 6 and 9). Concerning twinning, the "pole mechanism" proposed by Cottrell is designed to explain how a single dislocation can steadily traverse from one plane to its adjacent plane[36]. Although it successfully describes twinning in hexagonal close-packed materials[34], this mechanism, later recognized by Cottrell himself, fails to explain the progressive twin growth in fcc materials[13,37]. The mechanisms enabling the continuous nucleation of

twinning dislocations on successive planes in the bulk interior remain a topic of ongoing investigation[38,39]. Our simulation and experimental findings in studying E-SB provide insight into a dynamic source capable of producing partial dislocations on adjacent layers. As partial dislocation glides, it shifts the TB upwards by one atomic layer and concurrently relocates the source, ensuring a sustained process of twin growth. We find that this process of successive new source generation at the SF-TB junction, leading to E-SB formation, occurs in other fcc materials, such as Ag and Cu, but not in Al, in which full dislocation mechanism is dominant (Supplementary Fig. 20).

The two selected orientations in this study were chosen to activate different dislocation mechanisms and examine their correlation

with slip banding. In [100]-oriented crystals, dislocations predominantly operate as full, and partial dislocations prevail in [110]-oriented crystals. Despite equivalent Schmid factors across multiple slip planes (e.g., ABC and ABD planes in [100] crystals), variations in slip magnitudes can occur in real materials due to dislocation source heterogeneity. A higher source density on one plane promotes increased slip, leading to larger rotation angles and favoring slip concentration[25,40]. In the onset of plasticity, the initial Schmid factor symmetry begins to break down, giving rise to slip concentrating on one primary plane and manifesting as slip banding and strain localization. The formation of extended slip bands requires the activation of two non-parallel (non-coplanar) slip planes with partial dislocations. In contrast, if full dislocations are active, even under non-coplanar conditions, C-SBs would form, as shown in Supplementary Fig. 21a, b for the [112] orientation. For a single-slip system, such as the [123] orientation, the absence of interactions between non-parallel planes can lead to dominance of the Frank-Read mechanism, resulting in C-SB formation (Supplementary Fig. 21c, d). The single-arm source model, with one pinning point, has been proposed to explain the size effect on strength in nano- and micropillars[41,42]. However, when sample sizes exceed a few micrometers, this scaling relationship diminishes, indicating a shift from surface-controlled to bulk-dominated dislocation mechanisms[43]. While the single-arm source model explains size-dependent strength, its connection to slip band formation and evolution remains an open question. In nano-sized pillars, single-arm source operation may influence slip banding, but at larger scales, bulk dislocation sources are likely dominant. It is also noted that the current study focuses on more homogeneous material systems. The influence of precipitates, oxide nanoparticles, and their impact on dislocation mechanisms and resulting slip banding is an intriguing topic that deserves future investigation.

In summary, by employing in-situ mechanical compressions coupled with microstructural analyses and computational models, we unravel divergent evolution of slip banding in single-crystal CrCoNi alloys. Contrary to the longstanding Frank-Read model, our findings suggest that while the confined slip band originates from dislocation multiplication at an active source, the extended band emerges from partial dislocation glide-induced source deactivation and dynamic new source generation across succession planes. This study challenges the classic views of material deformation and strain localization in crystals and advances the fundamental understanding of atomic-scale dislocation motions and the resulting slip band evolution.

## Methods

### Materials and sample preparation
An equiatomic CrCoNi medium-entropy alloy (MEA) was fabricated using high-purity metals (purity ≥99.9%) by arc melting under a Ti-gettered high-purity argon atmosphere. The CrCoNi ingot was then sectioned into smaller specimens and divided into two groups. These samples were homogenized at 1250 °C for 12 h followed by rapid water quenching to obtain a homogeneous solid solution, referred to as the water-quenched alloy (Quenched). Subsequently, one group of the samples was subjected to further long-term aging at 925 °C for 120 h followed by furnace cooling, designated as the aged alloy (Aged). Prior to microstructural analysis, all samples were mechanically ground and electropolished using an electropolishing system (Electromet 4, Buehler, IL, USA).

### Micropillar preparation and in-situ compression testing
Electron backscattered diffraction (EBSD) analysis was performed to identify grain orientation using a scanning electron microscope (SEM, GAIA3, Tescan, Brno, Czech Republic) equipped with an EBSD system (AZtecHKL NordlysMax2, Oxford Industries Inc., GA, USA). According to the orientation mapping (Supplementary Fig. 1 and Fig. 16), we selected grains with orientations of [110] and [100] and then fabricated

single-crystalline micropillars within these grains using focused ion beam (FIB) in a FEI Quanta 3D Dual-Beam FIB/SEM. For each orientation, three cylindrical-shaped single-crystal micropillars, with dimensions of 5 × 10 μm, were prepared. A length-to-diameter aspect ratio of ~2 was maintained to avoid buckling due to higher aspect ratios and non-uniform stresses due to lower aspect ratios[44,45]. To minimize the contamination from Ga and tapering effect, the micropillars were milled using a voltage and current of 30 kV and 50 pA in the final FIB cleaning procedure. The micropillar design used in our study maintains a small taper angle of ~2°.

In-situ uniaxial compressive tests were performed at room temperature inside an SEM (Quanta 3D) using a Hysitron PI85 Picoindenter (Bruker Inc., Massachusetts, USA) equipped with a 20 μm diameter flat-top conical tip. The compression tests were conducted under displacement control mode with a displacement rate of 15 nm/s, corresponding to an initial strain rate of $10^{-3}$ s$^{-1}$.

### Slip band characterization from micrometer to atomic scales
The deformation behaviors and surface morphologies were first characterized using SEM (Supplementary Note 1). For microstructural investigation of slip bands at atomic resolution, transmission electron microscope (TEM) samples were fabricated from the post-deformed micropillar using the FIB lift-out method in an FEI Quanta 3D microscope equipped with an OmniProbe. A final polish at 5 kV and 48 pA was used to minimize the ion beam damage to the TEM sample. Ion beam damage on the TEM sample is further cleaned in a Fischione Nanomill™ Model 1040 system. Bright-field (BF)-Scanning/transmission electron microscopy (S/TEM) was used to examine the cross-sectional morphology and slip band characteristics inside a JEOL JEM-2800 TEM operated at 200 kV. High-angle annular dark field (HAADF)-STEM was used to characterize the atomic structures of slip bands in a JEOL JEM-ARM300F Grand ARM TEM with double Cs correctors operated at 300 kV. A probe current of 35 pA and inner and outer collection angles of 64 and 180 mrad were used for HAADF imaging.

### 4D-STEM measurement of deformation microstructure
In the four-dimensional (4D)-STEM experiment, we collect the entire convergent beam electron diffraction (CBED) patterns as 2D images for each scan position. This process, in conjunction with the 2D scanning grid, generates a 4D dataset. The 4D data were acquired on a JEOL JEM-ARM300F double-aberration-corrected TEM operating at 300 kV. The CBED patterns were captured using a Gatan OneView camera at a frame rate of 300 frames per second (fps), with each frame comprising 1024 × 1024 pixels. We employed a convergence semi-angle of 1 mrad using a 10 μm condenser (C4) and a camera length of 40 cm for the 4D-STEM measurements. To obtain high resolution for strain analysis, the slip band regions of samples were tilted using a standard double-tilt holder, ensuring that the desired [110] crystallographic axis is nearly parallel to the incident electron beam. The electron probe was raster-scanned across the selected area using a step size of 1.5 nm and the dwell time is 0.01 s for each step to minimize drift during pattern acquisition. Subsequently, the raw 4D-STEM data were subjected to machine and software binning, resulting in a final resolution of 256 × 256 pixel to enhance the signal-to-noise ratio for subsequent computational analysis. The data processing involved strain mapping was performed using scripts provided in the open-source py4DSTEM software package[46]. For example, Supplementary Fig. 8 shows 4D-STEM analysis of the extended slip band, revealing a large-size twin within E-SB.

### Intrinsic defects and Frank-Read source
During solidification under cooling or quenching from high temperatures, vacancies tend to condense, leading to the formation of a vacancy cluster, also known as a dislocation loop. In fcc materials, two predominant types of cluster exist, rhombic 1/2 < 110 > {110} perfect

loop and elliptical $1/3 < 111 > \{111\}$ Frank loop. We first determine the most stable loop shapes by comparing their formation energies (Supplementary Fig. 10). Specifically, perfect loop is created by removing atoms on (110) plane. By computing the formation energy as a function of loop aspect ratio and orientation, we identify the loop shape with lowest energy (Supplementary Fig. 10b). Similarly for the Frank loop, the circular shape is found to be the most stable geometry (Supplementary Fig. 10d).

Between the two types of loops, we find that only the rhombic perfect loop can act as a dislocation source, capable of repetitively emitting dislocations under mechanical stress. Supplementary Fig. 11a, b shows the perfect loop located on the (110) plane and its relative position with Thompson tetrahedron. The vertices of the rhombic loop serve as pinning points, and under shear stress, the dislocation can be nucleated. For example, the dislocation segment, BE of the loop, resides on the BCD slip plane and has the full Burgers vector of $1/2[110]$. As a result, the dislocation line, anchored at both ends, functions as an effective dislocation nucleation site (Frank-Read source) under shear stress. In contrast, the Frank loop, characterized by a Burgers vector of $1/3[111]$, lacks pinning points and cannot emit dislocations (Supplementary Fig. 11c, d). Instead, this loop typically undergoes one-dimensional migration under thermal activation, insensitive to mechanical stress[47].

## Atomistic modeling of slip banding
To elucidate the slip band formation mechanisms in CrCoNi MEAs, we conducted large-scale molecular dynamics (MDs) simulations of uniaxial compression. The face-centered cubic (fcc) single-crystal nanopillars were constructed with their loading axis ($z$ axis) oriented along [110] and [100] directions, respectively. These cylindrical-shaped single-crystal pillars, each containing 14 million atoms, have dimensions of 107 nm in height 43 nm in diameter. Dislocation loops were inserted into the pillars by carving out rhombic prismatic loops of the vacancy-type. Their Burgers vectors orientate along $1/2 < 110 >$, which is normal to the habit planes of loops[48], resulting in six distinct orientations of vacancy-type prismatic loops. The rhombus edges of these loops are aligned along two <112> directions perpendicular to the Burgers vector, situated on two {111} glide planes (Supplementary Fig. 11a, b). We uniformly distributed six distinct loops along the z-axis of the pillars and randomly placed them on the x-y planes. With the length of each rhombus dislocation loop around 55 Å, the initial dislocation density in the nanopillar is approximately $8.5 \times 10^{14}$ m$^{-2}$. A periodic boundary condition is applied to the z axis, while free surface conditions are set along x and y axes to demonstrate surface responses and impacts. Uniaxial compressive deformation was applied to the nanopillars at a constant engineering strain rate of $5 \times 10^7$ s$^{-1}$ along z-axis and a temperature of 300 K using a Langevin thermostat. A damping parameter for the Langevin thermostat is set to 5 ps to dissipate heat vibrations generated by dislocation motion. The atomistic simulations of deformation are carried out using the open-source Large-scale Atomic/Molecular Massively Parallel Simulator code[49]. The atomic structures are visualized using the Open Visualization Tool[50], and the dislocations are presented by the dislocation extraction algorithm[51].

## MC and MD simulation
We employed the hybrid Monte Carlo/Molecular Dynamics (MC/MD) simulation within the variance-constrained semigrand canonical ensemble (VCSGC) to construct the aged systems with chemical short-range order[52]. A total of 1 million MD time steps with a step size of 2.5 fs were carried out, and MC cycle was executed for every 100 time steps. During the MC simulations, we performed N/5 number of atom-type swap trails, where N is the total number of atoms in the system. On average, each atom is subjected to 2000 atom-type swap trails by the end of the simulation. The acceptance probability of atom-type swap

trails is determined by five parameters, including system energy change $\Delta u$ after a trail, concentration difference $\Delta c$ from the targeted equimolar concentration, chemical potential difference between two species $\Delta \mu$, variance parameter $k$, and system temperature $T$. We use the parameters $\Delta \mu_{Ni-Co} = 0.021$ and $\Delta \mu_{Ni-Cr} = -0.31$ eV derived from the semigrand canonical ensemble simulation[53], $k = 1000$, and $T = 650$ K, which ensure that the equimolar concentration is achieved by the end of simulation. In the MD intervals, the system stresses are relaxed to zero average value using Parrinello-Rahman barostat. It's noteworthy that the hybrid MC and MD simulation, incapable of modeling the diffusion kinetics associated with chemical order formation as in actual aging experiments, produces an equilibrated state determined by thermodynamics. Nevertheless, this approach provides a low-energy aged state, enabling us to study local chemical effects on deformation mechanisms.

## Deformation SF, hcp phase, and TB identification
Dislocation slip and plastic deformation of fcc materials can introduce a variety of atomic structures with hexagonal close-packed (hcp) coordination, manifested as intrinsic stacking fault, extrinsic stacking fault, twin boundary (TB), and hcp phase (lamella) in the material. The most frequently used structure characterization methods, such as common neighbor analysis[54] and centrosymmetry parameter analysis[55], which can effectively classify the local structure type (for instance, hcp) of each atom, are incompetent at differentiating different hcp-coordinated structures. Therefore, we define a weighted coordination number Z, measuring the number of same structure neighbors an atom has within a cutoff distance $r_c$, elaborated in our recent study[33]. When the cutoff distance is covering the two consecutive habit planes, i.e., $2.5a/\sqrt{3}$ (a is the lattice constant) in the middle of the second (111) and third (111) planes, all the hcp-coordinated structures, including SF, TB, and hcp phase, can be identified successfully from Z ($Z_{TB} = 18$, $Z_{iSF} = 24$, $Z_{eSF} = 30$, and $Z_{hcp} \geq 37$), as shown in Supplementary Fig. 12. The value of Z is calculated via coordination analysis of atoms within $r_c$ and with hcp structure, which is competent to capture the unique feature associated with each defect, realizing the structure identification.

## Data availability
All data generated for this study are included in the main text and its supplementary information, and Source Data. Other raw data related to this research are available from the corresponding author upon request. Source data are provided with this paper.

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

## Acknowledgements

The research was supported by the U.S. Department of Energy (DOE), Office of Basic Energy Sciences, under Award No. DE-SC0022295, and the start-up grant of P.C. from the University of California Irvine. H.C. was supported by the National Science Foundation (DMR–2105328). X.W. was partially supported by UC Irvine Center for Complex and Active Materials (DMR-2011967). The authors acknowledge the use of facilities and instrumentation at the UC Irvine Materials Research Institute (IMRI), supported in part by the National Science Foundation Materials Research Science and Engineering Center program through the UC Irvine Center for Complex and Active Materials (DMR-2011967). The authors thank T.Z. for the helpful discussion.

## Author contributions

P.C. advised the project and wrote the manuscript. B.J.X. designed the experiments and performed SEM and TEM characterizations. H.C. carried out atomistic simulations. P.C., B.J.X., and H.C. analyzed the data and presented the figures. C.Z. prepared the bulk samples, and M.X. and X.W. assisted in the TEM and SEM experiments. P.W., B.X., L.V., J.R., and X.P. contributed to the discussion of the results.

## Competing interests

The authors declare no competing interests.
