## [Transparent Peer Review file · Nature Communications]

Divergent Evolution of Slip Banding in CrCoNi Alloys

Corresponding Author: Professor Penghui Cao

Version 0:

Reviewer comments:

Reviewer #1

(Remarks to the Author)

Utilizing in-situ nanomechanical compressive tests, state of art characterization tools and atomistic simulation methods, this study offers a thorough examination of plastic deformation models across various orientations of CrCoNi alloys. The observed distinct deformation mechanism among different pillars are largely attributed to their unique deformation mechanisms. Specifically, two distinct dislocation sources are identified: Frank-Read dislocation source, predominantly governing deformation of [110] pillars by providing full dislocations, and the mobile dislocation source, mainly providing partial dislocations and controlling deformation of [100] pillars. Overall, the experimental and simulation investigations in this study are of high quality and provide novel insights into the plastic deformation model and strain localization in metallic materials. All the micrographs are very attractive. It can be considered for publication after addressing the following questions.

1. How to select a proper crystal orientation to enable the plastic deformation of pillars are controlled by full dislocations or partial dislocations?
2. The study identifies two distinct deformation mechanisms associated with pillars with different crystal orientations and proposes that the generation of partial dislocations by a mobile dislocation source, which plays a crucial role in the formation of planar defects. Is it possible that this phenomenon is just a consequence of the anisotropy in stacking fault energy in different orientation?
3. The fundamental mechanism outlined in this study suggests that the nucleation and expansion of planar defects resulting from the emergence of new dislocation sources at the front of deformation interface. Can you ascertain that these defects are initiated within the pillars interior rather than from the sample surface?
4. For the strain burst or uniform deformation observed in the compressed pillars with different orientation, how to identify which one is the dominant factor, twinning induced orientation variation for easy glide or the activation of twinning dislocation sources?

Reviewer #2

(Remarks to the Author)

The paper presents a consistent study explaining the formation of different types of slip bands (confined or extended) typically encountered in micropillar compression experiments. The results are supported both by very high-quality experimental and numerical observations. The results are of significance for the field of micromechanics. As a researcher with extensive experience performing micropillar compression experiments in a range of metallic materials, I have myself observed these intriguing phenomena, and it was clear to me that confined slip bands were associated to the operation of single dislocation sources and extended ones to the shutting down and activation of several sources. I have never come across a careful study that explains this, including the dislocation mechanisms that undergo in each case. For this, I sincerely congratulate the authors, since the experiments and simulations performed are of very high quality and the manuscript is very clearly presented. My major concern is that the manuscript enforces that the findings as universal, while I think they apply to this particular case, CrCoNi, and the particular orientations tested. In other material systems, there might be other mechanisms shutting down sources and activating new ones in adjacent planes (i.e., intersection with other slip

bands, precipitates, other dislocations, etc...) that can lead to diffuse slip.

Related to this, I would appreciate some clarifications from the authors:

1. I think the authors overdue the wide impact of their findings in the abstract and the last paragraph of the conclusion section. I clearly see the impact for the micropillar compression community, but beyond that, I do not think the findings have a large impact on other research topics, since slip avalanches and intermittent flow are (almost) limited to the deformation of micrometer scale volumes. I do not think either that diffuse slip bands only occur by the mechanism proposed. The results present strong evidence to support that the mechanism applies in the [001] oriented CrCoNi micropillars, but there might be other mechanisms that shut down dislocation sources in other material systems, or even in the same material, in other orientations.

2. The observations are limited to two different micropillar compression orientations: [011] and [001]. However, the authors derive general conclusions based on only these 2 orientations, and this raises several questions.

2.1. In both, multiple slip systems are activated according to the SF analysis: 4 (in 2 planes) for [001] and 8 (in 4 planes) for [001]. Indeed, the experimental observations show the activation of several slip systems, but in all cases, one predominates (the one the authors call the primary slip system). They explain this by rotation, but how do they measure rotation? Actually, the rotation is a consequence of the activation of this predominant slip system. Is this an assumption? Was there a small misorientation on the initial micropillar orientation that favored one slip system over the others? If the micropillars were oriented exactly along [011] or [001] there is no reason why one slip system must be predominant over the others.

2.2 In both orientations, there are several co-planar and non co-planar slip systems acting simultaneously. Does this influence the conclusions? i.e., what would happen in orientations with multiple slip-systems that are only co-planar or only not co-planar? And what about a single slip orientation like [123]?

2.3. Additionally, the proposed mechanism for the shift of the dislocation source in the diffuse slip band requires the intersection of the TB and a SF in the secondary slip plane. Does that imply that this mechanism would not operate in a single slip oriented pillar of CrCoNi? Even if the SF is bigger in the leading partial and this escapes the surface of the pillar leaving the trailing partial behind? Could you observe diffuse slip in that case? How would the dislocation source migrate in the absence of SFs in the secondary slip system?

3. The authors only consider three possible nucleation stresses: perfect F-R source, surface step and perfect pillar in Supplementary Figures 6 and 9. The accepted deformation mechanism in micropillars is the activation and operation of single-arm dislocation sources (i.e., one pinning point inside the pillar and the other on the surface), which explains micropillar size effects. However, this is not considered, at least explicitly (is it implied in the size of the F-R source?) Please comment. Additionally, how these nucleation stresses are computed should also be added on the supplementary material section.

In summary, a very complete and convincing experimental and simulation study explaining why and how confined slip occurs in CrCoNi micropillars oriented close to [011] and extended slip occurs in CrCoNi micropillars oriented close to [001]. However, the significance and universality of these observations to extend these conclusions to other orientations and/or other material systems is not convincingly demonstrated.

Version 1:

Reviewer comments:

Reviewer #1

(Remarks to the Author)

This manuscript presents a systematic investigation that advances our fundamental understanding of slip band formation mechanisms in crystalline materials. Through micropillar compression experiments, the authors have successfully elucidated the distinct formation pathways of confined versus extended slip bands. More importantly, the work introduces groundbreaking defect-mediated deformation mechanisms that operate under varying crystallographic orientations. The revised version significantly strengthens these claims through comprehensive experimental validation and mechanistic analysis, establishing a reliable framework for understanding microstructural evolution. These findings not only provide transformative insights into the atomic-scale processes governing plastic deformation but also establish a new paradigm for controlling strain localization in metallic systems. The quality and impact of this work meet the high standards expected for publication in Nature Communications.

Reviewer #2

(Remarks to the Author)

I thank the authors for addressing my concerns. The paper presents a consistent study explaining the formation of different slip bands (confined and extended) in CrCoNi micropillars, supported by very high quality experimental and numerical observations. I enjoyed reading the revised manuscript and the supplementary information provided. Good work!!

Reply to reviewers:

The comments from the reviewers are presented in black, with our responses highlighted in blue. Revisions in the Manuscript and Supplementary Information (SI) are highlighted by red color, and a summary of the changes is provided at the end of this document.

Reviewer #1:

Utilizing in-situ nanomechanical compressive tests, state of art characterization tools and atomistic simulation methods, this study offers a thorough examination of plastic deformation models across various orientations of CrCoNi alloys. The observed distinct deformation mechanism among different pillars are largely attributed to their unique deformation mechanisms. Specifically, two distinct dislocation sources are identified: Frank-Read dislocation source, predominantly governing deformation of [110] pillars by providing full dislocations, and the mobile dislocation source, mainly providing partial dislocations and controlling deformation of [100] pillars. Overall, the experimental and simulation investigations in this study are of high quality and provide novel insights into the plastic deformation model and strain localization in metallic materials. All the micrographs are very attractive. It can be considered for publication after addressing the following questions.

Reply: We thank the reviewer very much for the supportive remarks that “*experimental and simulation investigations in this study are of high quality and provide novel insights*”. We have provided detailed responses to each of your questions below.

1. How to select a proper crystal orientation to enable the plastic deformation of pillars are controlled by full dislocations or partial dislocations?

Reply: The selection of crystal orientation to activate full or partial dislocations is based on the Schmid factors of leading partial (LP) and trailing partial (TP) dislocations for different loading orientations.

For example, when compressing along [110] orientation, two of the four slip planes, specifically (111) and ($\bar{1}\bar{1}\bar{1}$), (i.e., ABC and ABD in Thompson tetrahedron notation) will be activated due to their largest Schmid factor, as shown in Table R1. On the ABC plane, the Schmid factor of TP is 0.47, which is nearly double that of the LP Schmid factor of 0.2357 (last two columns of Table R1). This indicates that under compression, a larger driving force is acting on TP than that of LP. Even if the stacking fault energy is low, the dislocations tend to move in pairs – a full dislocation mechanism. The process of emitting a series of full dislocations is further illustrated in Fig. 4a of the Manuscript.

In contrast, when selecting the compression direction along [100], the LP dislocation has a much larger Schmid factor than the TP (Table R2). Considering the low stacking fault energy, the lower driving force acting on the TP dislocation can be insufficient to drive its motion. Therefore, dislocations in this orientation primarily operate in the form of leading partial, not in pairs (full dislocation).

Table R1. Schmid factors of slip systems for [110]-oriented crystal.

Slip plane	Full dislocations		Partial dislocations	
	Slip direction	Schmid factor	Slip direction	Schmid factor
ABC (111)	AC $[01\bar{1}]$	0.4082	LP $A\delta [\bar{1}2\bar{1}]$	0.2357
			TP $\delta C [11\bar{2}]$	0.4714
	BC $[10\bar{1}]$	0.4082	LP $B\delta [2\bar{1}\bar{1}]$	0.2357
			TP $\delta C [11\bar{2}]$	0.4714
	AB $[\bar{1}10]$	0	LP $A\delta [\bar{1}2\bar{1}]$	0.2357
			TP $\delta B [\bar{2}11]$	-0.2357
ABD ($\bar{1}\bar{1}1$)	AD $[\bar{1}0\bar{1}]$	0.4082	LP $A\gamma [\bar{2}1\bar{1}]$	0.2357
			TP $\gamma D [11\bar{2}]$	0.4714
	BD $[0\bar{1}\bar{1}]$	0.4082	LP $B\gamma [12\bar{1}]$	0.2357
			TP $\gamma D [11\bar{2}]$	0.4714
	AB $[\bar{1}10]$	0	LP $A\gamma [\bar{2}1\bar{1}]$	0.2357
			TP $\gamma B [\bar{1}21]$	-0.2357
ACD ($\bar{1}\bar{1}\bar{1}$)	AD $[\bar{1}0\bar{1}]$	0	LP $A\beta [\bar{1}1\bar{2}]$	0
			TP $\beta D [\bar{2}1\bar{1}]$	0
	AC $[01\bar{1}]$	0	LP $A\beta [\bar{1}1\bar{2}]$	0
			TP $\beta C [12\bar{1}]$	0
	CD $[\bar{1}\bar{1}0]$	0	LP $C\beta [\bar{1}\bar{2}1]$	0
			TP $\beta D [\bar{2}1\bar{1}]$	0
BCD ($\bar{1}1\bar{1}$)	BC $[10\bar{1}]$	0	LP $B\alpha [1\bar{1}\bar{2}]$	0
			TP $\alpha C [21\bar{1}]$	0
	BD $[0\bar{1}\bar{1}]$	0	LP $B\alpha [1\bar{1}\bar{2}]$	0
			TP $\alpha D [\bar{1}\bar{2}\bar{1}]$	0
	CD $[\bar{1}\bar{1}0]$	0	LP $C\alpha [\bar{2}\bar{1}1]$	0
			TP $\alpha D [\bar{1}\bar{2}\bar{1}]$	0

Table R2. Schmid factors of slip systems for [100]-oriented crystal.

Slip plane	Full dislocations		Partial dislocations	
	Slip direction	Schmid	Slip direction	Schmid factor
ABC (111)	BA $[\bar{1}\bar{1}0]$	0.4082	LP B δ $[\bar{2}\bar{1}\bar{1}]$	0.4714
			TP δ A $[\bar{1}\bar{2}\bar{1}]$	0.2357
	BC $[10\bar{1}]$	0.4082	LP B δ $[\bar{2}\bar{1}\bar{1}]$	0.4714
			TP δ C $[\bar{1}\bar{1}\bar{2}]$	0.2357
	AC $[01\bar{1}]$	0	LP A δ $[\bar{1}\bar{2}\bar{1}]$	-0.2357
			TP δ C $[\bar{1}\bar{1}\bar{2}]$	0.2357
ABD ($\bar{1}\bar{1}\bar{1}$)	AD $[\bar{1}0\bar{1}]$	0.4082	LP A γ $[\bar{2}\bar{1}\bar{1}]$	0.4714
			TP γ D $[\bar{1}\bar{1}\bar{2}]$	0.2357
	AB $[\bar{1}\bar{1}0]$	0.4082	LP A γ $[\bar{2}\bar{1}\bar{1}]$	0.4714
			TP γ B $[\bar{1}\bar{2}\bar{1}]$	0.2357
	DB $[011]$	0	LP D γ $[\bar{1}\bar{1}\bar{2}]$	-0.2357
			TP γ B $[\bar{1}\bar{2}\bar{1}]$	0.2357
ACD ($1\bar{1}\bar{1}$)	DA $[101]$	0.4082	LP D β $[\bar{2}\bar{1}\bar{1}]$	0.4714
			TP β A $[\bar{1}\bar{1}\bar{2}]$	0.2357
	DC $[110]$	0.4082	LP D β $[\bar{2}\bar{1}\bar{1}]$	0.4714
			TP β C $[\bar{1}\bar{2}\bar{1}]$	0.2357
	CA $[0\bar{1}1]$	0	LP C β $[\bar{1}\bar{2}\bar{1}]$	-0.2357
			TP β A $[\bar{1}\bar{1}\bar{2}]$	0.2357
BCD ($\bar{1}\bar{1}\bar{1}$)	CB $[\bar{1}01]$	0.4082	LP C α $[\bar{2}\bar{1}\bar{1}]$	0.4714
			TP α B $[\bar{1}\bar{1}\bar{2}]$	0.2357
	CD $[\bar{1}\bar{1}0]$	0.4082	LP C α $[\bar{2}\bar{1}\bar{1}]$	0.4714
			TP α D $[\bar{1}\bar{2}\bar{1}]$	0.2357
	BD $[0\bar{1}\bar{1}]$	0	LP B α $[\bar{1}\bar{1}\bar{2}]$	-0.2357
			TP α D $[\bar{1}\bar{2}\bar{1}]$	0.2357

2. The study identifies two distinct deformation mechanisms associated with pillars with different crystal orientations and proposes that the generation of partial dislocations by a mobile dislocation source, which plays a crucial role in the formation of planar defects. Is it possible that this phenomenon is just a consequence of the anisotropy in stacking fault energy in different orientation?

Reply: The face-centered cubic (FCC) structure has slip systems that operate on four non-parallel $\{111\}$ planes ($\langle 112 \rangle$ directions for partial dislocations and $\langle 110 \rangle$ directions for full dislocation). The stacking fault energy (SFE) in FCC crystals is related to disruptions in the stacking sequence on the $\{111\}$ planes. The SFE remains uniform across all slip planes under different loading orientations, as these structures exhibit slip isotropy.

The reviewer's question is critical for anisotropic crystals, such as hexagonal close-packed (HCP) metals, where the SFE is nonuniform across basal, prismatic, or pyramidal planes. In such cases, anisotropy in SFE can play a major role in determining the deformation mechanism.

In FCC structures, however, the dislocation experiences the same SFE across all $\{111\}$ planes due to slip isotropy. While the SFE itself may not vary with crystal orientation in FCC structure, the loading direction significantly affects the Schmid factors of leading and trailing partial dislocations. These Schmid factors determine whether the trailing partial dislocation can be activated, which influences the formation ability of planar defects.

3. The fundamental mechanism outlined in this study suggests that the nucleation and expansion of planar defects resulting from the emergence of new dislocation sources at the front of deformation interface. Can you ascertain that these defects are initiated within the pillars interior rather than from the sample surface?

Reply: To determine whether dislocation nucleation originates from the sample surface or within the pillar interior, we performed atomistic modeling and compared the dislocation nucleation stresses for three scenarios: free surface, surface steps, and the pillar interior. The results, presented in Fig. R1 (also shown as Supplementary Fig. 9 in the Supplementary Information), reveal that the stress required to activate dislocations is lowest for pillars containing an interior defect source. Specifically, when the pinning point distance in the Frank-Read source increases from 3 to 6 and 9 nm, the nucleation stress decreases significantly from approximately 1.7 to 0.9 and 0.5 GPa, respectively. In contrast, the activation stress for a pillar with a free surface is notably higher, around 2.5 GPa. These findings suggest that dislocations are more likely to nucleate from the bulk interior, where the nucleation stress is substantially lower compared to that required for surface nucleation, even in the presence of surface steps.

To address this important point, we have added a new section to the amended Supplementary Information, titled “*Supplementary Note 4 | Dislocation nucleation stress from Frank-Read source, surface step, and surface*”, to clarify these results. We thank the reviewer for highlighting this important aspect.

Fig. R1. Dislocation nucleation stress comparison at Frank-Read source, surface step, and free surface, for loading in [100] direction. (a) Variation of yield stress with pinning point distance (F-R source), surface step size, and pillar diameter. (b) Corresponding structures post-dislocation emission (i.e., after yielding). For pillars with Frank-Read source, only non-FCC atoms are displayed to show the slip initiated from sample interior.

4. For the strain burst or uniform deformation observed in the compressed pillars with different orientation, how to identify which one is the dominant factor, twinning induced orientation variation for easy glide or the activation of twinning dislocation sources?

Reply: During the plastic flow stage of compressed pillars, distinct differences in strain bursts and stress drops are observed depending on the orientation. In the [110]-oriented micropillar with confined slip bands (Supplementary Fig. 2), evident strain bursts are accompanied by large stress drops. In contrast, in the [100]-oriented micropillar with extended slip bands, these ultra-large stresses drop events are significantly diminished, resulting in more uniform deformation (Supplementary Fig. 7).

In confined slip bands, strain bursts are driven by Frank-Read dislocation sources, which generate dislocations at fixed sites. Dislocation loops accumulate back stress as they expand, temporarily halting generation until stress reaches a critical level, leading to strain bursts and significant stress drops. In contrast, extended slip bands are governed by twinning dislocations, which glide across planes, promoting continuous twin growth and uniform deformation. This process involves

relocating dislocation sources after emitting loops, enabling smooth and continuous dislocation generation, which lowers strain bursts and stress drops.

Reviewer #2:

The paper presents a consistent study explaining the formation of different types of slip bands (confined or extended) typically encountered in micropillar compression experiments. The results are supported both by very high-quality experimental and numerical observations. The results are of significance for the field of micromechanics. As a researcher with extensive experience performing micropillar compression experiments in a range of metallic materials, I have myself observed these intriguing phenomena, and it was clear to me that confined slip bands were associated to the operation of single dislocation sources and extended ones to the shutting down and activation of several sources. I have never come across a careful study that explains this, including the dislocation mechanisms that undergo in each case. For this, I sincerely congratulate the authors, since the experiments and simulations performed are of very high quality and the manuscript is very clearly presented. My major concern is that the manuscript enforces that the findings as universal, while I think they apply to this particular case, CrCoNi, and the particular orientations tested. In other material systems, there might be other mechanisms shutting down sources and activating new ones in adjacent planes (i.e., intersection with other slip bands, precipitates, other dislocations, etc...) that can lead to diffuse slip.

Reply: The authors thank the reviewer for the constructive and supportive comments and for providing valuable context on this topic. We are also grateful for the encouraging notes from an expert in the field regarding the quality and significance of our work.

It is noted that, to explore the applicability of our proposed mechanisms in other material systems, we performed atomistic simulations on Cu, Ag, and Al. The results of these simulations are summarized in Supplementary Fig. 20 of the Supplementary Information. Even with these findings, we agree that our study does not support “the finding is universal”. To address the reviewer’s concern about the potential overstatement in the findings, we have revised the manuscript to explicitly specify that our study is focused on CrCoNi alloys. These clarifications have been included in both the abstract and the conclusion section to ensure there is no ambiguity for future readers. The current study focuses on more homogeneous material systems, yet the presence of precipitates and its impact on diffuse slip band, raised by the reviewer, is very intriguing and certainly deserves another careful study.

Related to this, I would appreciate some clarifications from the authors:

1. I think the authors overdue the wide impact of their findings in the abstract and the last paragraph of the conclusion section. I clearly see the impact for the micropillar compression community, but beyond that, I do not think the findings have a large impact on other research topics, since slip avalanches and intermittent flow are (almost) limited to the deformation of micrometer scale volumes. I do not think either that diffuse slip bands only occur by the mechanism proposed. The results present strong evidence to support that the mechanism applies in the [001] oriented CrCoNi

micropillars, but there might be other mechanisms that shut down dislocation sources in other material systems, or even in the same material, in other orientations.

Reply: We thank the reviewer for sharing the feedback on the abstract and the last paragraph and for highlighting the potential overstatement of the broader impact. We consider the primary focus of the study is on revealing the (full and partial) dislocation mechanisms governing the formation and evolution of two distinct slip bands. Slip avalanches and intermittent flow are results of dislocation dynamics and their interaction with defects. In the manuscript, we discussed the impact of the different dislocation mechanisms and slip banding on slip avalanche and intermittent plastic flow, rather than making generalized claims about slip avalanche. To address the reviewer's concerns and ensure the manuscript more accurately reflects the study's scope, we have revised the abstract and conclusion, particularly removed the statement "deformation instability, slip avalanche, twinning mechanisms ..." from paragraph. The changes are highlighted in red text of the manuscript and also summarized at the end of the response letter.

2. The observations are limited to two different micropillar compression orientations: [011] and [001]. However, the authors derive general conclusions based on only these 2 orientations, and this raises several questions.

Reply: The selection of two orientations are intentional to active different dislocation mechanisms. For [110] orientation, the Schmid factor of trailing partial is nearly double that of its corresponding leading partial dislocation (the Schmid factors summarized in Table R1 and Table 1 of Supplementary Information). Based on this, we hypothesized and later confirmed (in Fig. 1d and Fig. 3a of Manuscript) that dislocations predominantly operate in the form of full character in this orientation. In contrast, the [100] orientation was selected to trigger the partial dislocation mechanism, as its Schmid factor is significantly larger for the leading partial compared to the trailing partial (Table R2 and Table 2 of Supplementary Information). The operation of partial dislocation mechanism is demonstrated in our experiments (Fig. 2e-f, Fig. 5e) and simulations (Fig. 3b, Fig. 5d). We are responding the related questions in detail below.

2.1. In both, multiple slip systems are activated according to the SF analysis: 4 (in 2 planes) for [001] and 8 (in 4 planes) for [001]. Indeed, the experimental observations show the activation of several slip systems, but in all cases, one predominates (the one the authors call the primary slip system). They explain this by rotation, but how do they measure rotation? Actually, the rotation is a consequence of the activation of this predominant slip system. Is this an assumption? Was there a small misorientation on the initial micropillar orientation that favored one slip system over the others? If the micropillars were oriented exactly along [011] or [001] there is no reason why one slip system must be predominant over the others.

Reply: The Schmid factor, reflecting the resolve shear stress acting on the slip system, has symmetry, for example, in [110] or [100]-oriented pillars. Like in the [110] orientation, resolved shear stress on ABC and ABD planes are equivalent. However, the actual slip magnitudes on the slip plane in real materials can vary, depending on the initial dislocation source density. If the sources density associated with one slip plane is higher than the other, more slip would occur in such plane. This larger amount of slip will result in a higher rotation angle associated with this

plane. This rotation will further increase the Schmid factor of this runaway system and decrease the Schmid factor of the other competing system (as shown in Fig. R2b). Consequently, the initial Schmid factor symmetry breaks down, and one of slip plane becomes predominated.

Fig. R2. Slip-induced rotation breaking the initial Schmid factor symmetry in the [110]-oriented pillar. (a-b) Schematic illustration of crystal rotation due to slip under uniaxial compression. (c) The initial Schmid factor symmetry is breaking due to slip magnitude anisotropy. The rotation causes the ABC plane to become the predominated slip system.

Fig. R3. Atomistic modeling of slip-induced crystal rotation behavior. (a-c) The orientation of slip plane ABC changes due to the formation of slip banding.

To demonstrate this in detail, we present the atomistic modeling results for [110]-oriented pillar (Fig. R3). Because of dislocation loops (sources) heterogeneity, we find that the actual slip magnitude along each plane can be different. Fig. R3 shows that one of the two slip planes, ABC, carries more slip than the other one. Hence, a rotation occurs due to ABC slip. This rotation can further promote the Schmid factor on the ABC plane (right panel of Fig. R2c). Therefore, even for a micropillar with planes that have exactly the same Schmid factors, slip magnitude along each plane can be distinct. The initial symmetry in the Schmid factor, which remains valid until the onset of plasticity, breaks down in the plastic deformation stage because of rotation^{1,2}. Slip concentrates on one (primary) plane, manifesting as slip banding and strain localization. To clarify this important point, we have revised the manuscript and incorporated the discussion.

2.2 In both orientations, there are several co-planar and non co-planar slip systems acting simultaneously. Does this influence the conclusions? i.e., what would happen in orientations with multiple slip-systems that are only co-planar or only not co-planar? And what about a single slip orientation like [123]?

Reply: The introduced mechanism driving the extended slip band formation is dynamic dislocation sources creation at the intersections of stacking fault and twin boundary (Fig. 5b-e of Manuscript). This requires activation of two non-parallel slip planes, and more importantly, on which partial dislocation is operating. If full dislocation mechanism is active, under non-coplanar slip condition (two non-parallel slip plane), it is reasonable to speculate that a confined slip band rather than extended band would form. We present the results to verify this.

For [112] orientation (non-coplanar), there are two slip systems, involving two planes (ACD and BCD) with two non-coplanar slip systems. The Schmid factors of the trailing partials are consistently larger than those of the leading partials (seeing Table R3 below), suggesting a primary full dislocation mechanism. We performed the same uniaxial compression modeling, and the results are show in Fig. R4. It can be seen that, strain concentrated slip band (C-SB) is formed, instead of extended slip band (E-SB).

For pillar with single slip system or co-planar slip, the repetitive generation of partial dislocation sources will likely not occur due to the lack of the reaction between partial dislocation and twin boundary on two non-parallel planes. To demonstrate it, we prepare a pillar with [123] orientation that contains one slip system, and confined slip band forms along ACD plane (Table R4 and Fig. R5).

The authors thank the reviewer for bringing these questions. We have incorporated these results in the amended Manuscript and Supplementary Information.

Table R3. Schmid factors for the two active slip planes, ACD and BCD

Slip plane	Full dislocations		Partial dislocations	
	Slip direction	Schmid factor	Slip direction	Schmid factor
ACD ($\bar{1}\bar{1}\bar{1}$)	AD [$\bar{1}0\bar{1}$]	0.4082	LP A β [$\bar{1}\bar{1}\bar{2}$]	0.3143
			TP β D [$\bar{2}\bar{1}\bar{1}$]	0.3928
	AC [$01\bar{1}$]	0.1361	LP A β [$\bar{1}\bar{1}\bar{2}$]	0.3143
			TP β C [$12\bar{1}$]	-0.0786
	CD [$\bar{1}\bar{1}0$]	0.2722	LP C β [$\bar{1}\bar{2}1$]	0.0786
			TP β D [$\bar{2}\bar{1}\bar{1}$]	0.3928
BCD ($\bar{1}\bar{1}\bar{1}$)	BD [$0\bar{1}\bar{1}$]	0.4082	LP B α [$\bar{1}\bar{1}\bar{2}$]	0.3143
			TP α D [$\bar{1}\bar{2}1$]	0.3928
	BC [$10\bar{1}$]	0.1361	LP B α [$\bar{1}\bar{1}\bar{2}$]	0.3143
			TP α C [$21\bar{1}$]	-0.0786
	CD [$\bar{1}\bar{1}0$]	0.2722	LP C α [$\bar{2}\bar{1}\bar{1}$]	0.0786
			TP α D [$\bar{1}\bar{2}\bar{1}$]	0.3928

Fig. R4. Deformation behavior and slip banding in $[112]$ -oriented pillar. (a) The system has two activated slip planes (non-coplanar) with primary full dislocation mechanism. (b) Strain map and deformation structure show the formation of a confined slip band (C-SB).

Table R4. Schmid factors for the one active system on ACD plane

Full dislocations			Partial dislocations	
Slip plane	Slip direction	Schmid factor	Slip direction	Schmid factor
ACD ($1\bar{1}\bar{1}$)	AD [$\bar{1}0\bar{1}$]	0.4666	LP A β [$\bar{1}1\bar{2}$]	0.3367
			TP β D [$\bar{2}1\bar{1}$]	0.4714
	AC [$01\bar{1}$]	0.1166	LP A β [$\bar{1}1\bar{2}$]	0.3367
			TP β C [$12\bar{1}$]	-0.1347
	CD [$\bar{1}\bar{1}0$]	0.3499	LP C β [$\bar{1}\bar{2}1$]	0.1347
			TP β D [$\bar{2}\bar{1}\bar{1}$]	0.4714

Fig. R5. Confined slip band (C-SB) formation in $[123]$ -oriented pillar. (a) The system has one activate slip system on ACD plane with primary full dislocation mechanism. (b) Strain map and deformation structure show formation of C-SB.

2.3. Additionally, the proposed mechanism for the shift of the dislocation source in the diffuse slip band requires the intersection of the TB and a SF in the secondary slip plane. Does that imply that this mechanism would not operate in a single slip oriented pillar of CrCoNi? Even if the SF is bigger in the leading partial and this escapes the surface of the pillar leaving the trailing partial behind? Could you observe diffuse slip in that case? How would the dislocation source migrate in the absence of SFs in the secondary slip system?

Reply: In our response to the previous question, our proposed mechanism for extended slip band is not activated in pillars with a single slip system. In the case of $[123]$ orientation, the Schmid

factor of trailing partial is larger than the leading one, and dislocation can operate in the full character. Consequently, a confined slip band forms as a result of the Frank-Read mechanism.

The reviewer raised an interesting point about the slip banding in single slip system, where the Schmid factor for the leading partial is larger. In this case, a larger force would act on the leading partial while a smaller force acts on the trailing partial. During the onset of yielding, if only the leading partial is active, its annihilation at the surface would leave an extended stacking fault and the trailing partial behind. As the applied strain on the pillar continues to increase, the stress could eventually become high enough to activate the trailing partial. The glide of the trailing dislocation would remove the stacking fault and restore the FCC structure on the slip plane. Once this occurs, a new leading partial would be emitted from the Frank-Read source, initiating a repetitive process that could result in the formation of a confined slip band. This mechanism aligns with the process depicted in Fig. 4a of the manuscript, although the separation between the leading and trailing partials would likely be significantly larger in this scenario.

3. The authors only consider three possible nucleation stresses: perfect F-R source, surface step and perfect pillar in Supplementary Figures 6 and 9. The accepted deformation mechanism in micropillars is the activation and operation of single-arm dislocation sources (i.e., one pinning point inside the pillar and the other on the surface), which explains micropillar size effects. However, this is not considered, at least explicitly (is it implied in the size of the F-R source?) Please comment. Additionally, how these nucleation stresses are computed should also be added on the supplementary material section.

Reply: As the reviewer highlighted, the single-arm source (SAS) theory was proposed to explain the size effect on strength in micropillars. The scaling relationship between strength and pillar diameter diminishes when the sample size exceeds a few micrometers, as noted in prior studies related to SAS^{3,4}. This implies that the predominated dislocation mechanism is shifting from surface-controlled (like SAS) to bulk-dictated. For the 5-micrometer pillars in this study, the formation of slip bands with lengths greater than 10 micrometers is likely mediated by bulk dislocation mechanisms. In our atomistic models, dislocation sources were randomly introduced into the pillars (as described in the Methods section of the manuscript). The activation of these sources was not pre-defined but emerged naturally, based on their activation stresses determined by the atomistic configuration. For a Frank-Read source located near the surface, it could potentially operate with a single pinning point, if it exhibits a lower activation stress.

It is noted that, the well-known Frank-Read model, originally proposed to describe slip band formation, is central to this study's focus. While the SAS model nicely explains the size-dependent strength, its connection to slip band formation remains a question. In the nano-sized pillar, the operation of SAS might impact the slip band formation. At the bulk scale, bulk dislocation sources with two pinning points are likely dominant.

In response to this comment, we have also added the Supplementary Note 4, “Dislocation nucleation stress from Frank-Read source, surface step, and surface” to the revised Supplementary Information. We provide a detailed specification of the dislocation nucleation stresses for the three sources. The Supplementary Note 4 is provided below to facilitate reading.

Supplementary Note 4 | Dislocation nucleation stress from Frank-Read source, surface step, and surface

To determine whether dislocation nucleation originates from the sample surface or within the pillar interior, we performed atomistic modeling to compare the dislocation nucleation stresses across three scenarios: free surface, surface step, and Frank-Read source.

We created three perfect pillars with diameters of 22 nm, 32 nm, and 42 nm, each with an aspect ratio of 2.5, to study dislocation nucleation stress at the free surface. These pillars contain approximately 2 million, 6 million, and 14 million atoms, respectively. To calculate the nucleation stress in the presence of a surface step, we used a perfect pillar with a 32 nm diameter and introduced surface steps. Three step sizes were generated by shifting the pillar above the (111) slip plane along the [110] direction by a few Burgers vectors. The ratios of surface step size to pillar diameter, 0.01, 0.04, and 0.09, are considered. For dislocation nucleation stress at different sized Frank-Read sources, we inserted dislocation sources with varying pinning distances, specifically 3, 6, and 9 nm. For the three groups of system, uniaxial compressive deformation is applied to the nanopillars at a constant engineering strain rate of $5 \times 10^7 \text{ s}^{-1}$ along z-axis and a temperature of 300 K using a Langevin thermostat (Methods). The yielding stresses, at which the dislocation source is activated and plastic deformation occurs, were extracted and are presented in Supplementary Fig. 6 (for [110] orientation) and Fig. 9 ([110] orientation).

For example, the results shown in Supplementary Fig. 9 reveal that the stress required to activate dislocations is lowest for pillars containing an interior defect source. Specifically, when the pinning point distance in the Frank-Read source increases from 3 to 6 and 9 nm, the nucleation stress decreases significantly from approximately 1.7 to 0.9 and 0.5 GPa, respectively. In contrast, the activation stress for a pillar with a free surface is notably higher, around 2.5 GPa. These findings suggest that dislocations are more likely to nucleate from the bulk interior at the source, where the nucleation stress is substantially lower compared to that required for surface nucleation, even in the presence of surface steps.

In summary, a very complete and convincing experimental and simulation study explaining why and how confined slip occurs in CrCoNi micropillars oriented close to [011] and extended slip occurs in CrCoNi micropillars oriented close to [001]. However, the significance and universality of these observations to extend these conclusions to other orientations and/or other material systems is not convincingly demonstrated.

Reply: We thank the reviewer for providing valuable feedbacks on the manuscript. While we believe the dislocation mechanisms introduced may provide insights and implications for other FCC systems, we acknowledge further investigations are necessary to demonstrate “universality”. We have toned down the universality and significance in the revised abstract and conclusion section and presented the findings in a more focused manner.

Revision and changes to the manuscript and SI

Major revisions

- Abstract (page 1):

leading to rapid band thickening. Our findings provide ~~critical~~new insights into atomic-scale collective dislocation motion and microscopic deformation instability in advanced structural materials, ~~marking a pivotal advancement in our fundamental understanding of deformation dynamics.~~

- Summary (page 13):

planes. Our study challenges the classic views of material deformation and strain localization in crystals and advances the fundamental understanding of atomic-scale dislocation motions and the resulting slip band evolution, ~~having an impact on many research topics, including deformation instability, slip avalanche, twinning mechanisms, and tailoring of material properties through microstructural engineering.~~

- Adding discussion on Schmid factor symmetry and crystal orientation (page 12)

The two selected orientations in this study were chosen to activate different dislocation mechanisms and examine their correlation with slip banding. In [100]-oriented crystals, dislocations predominantly operate as full dislocations, and partial dislocations prevail in [110]-oriented crystals. Despite equivalent Schmid factors across multiple slip planes (e.g., ABC and ABD planes in [100] crystals), variations in slip magnitudes can occur in real materials due to dislocation source heterogeneity. A higher source density on one plane promotes increased slip, leading to larger rotation angles and favoring slip concentration^{25,40}. In the onset of plasticity, the initial Schmid factor symmetry begins to break down, giving rise to slip concentrating on one primary plane and manifesting as slip banding and strain localization. The formation of extended slip bands E-SBs requires activation of two non-parallel (non-coplanar) slip planes with partial dislocations. In contrast, if full dislocations are active, even under non-coplanar conditions, C-SBs would form, as shown in Supplementary Fig. 21a-b for the [112] orientation. For single-slip system, such as the [123] orientation, the absence of interactions between non-parallel planes can lead to dominance of the Frank-Read mechanism, resulting in C-SB formation (Supplementary Fig. 21c-d). The single-arm source mechanism, with one pinning point, has been proposed to explain the size effect on strength in nano- and micro-pillars^{41,42}. However, when sample sizes exceed a few micrometers, this scaling relationship diminishes, indicating a shift from surface-controlled to bulk-dominated dislocation mechanisms⁴³. While the single-arm source model explains size-dependent strength, its connection to slip band formation and evolution remains an open question. In nano-sized pillars, single-arm source operation may influence slip banding, but at larger scales, bulk dislocation sources are likely dominant. It is also noted that the current study focuses on more homogeneous material systems. The influence of precipitates, oxide nanoparticles, and their impact on dislocation mechanisms and resulting slip banding is an intriguing topic that deserves future investigation.

- Slip banding in non-coplanar and single slip system (page 22 of SI)

Supplementary Fig. 21 | Deformation behavior and slip banding in [112]-oriented and [123]-oriented pillars. (a) The system has two activated slip planes (non-coplanar) with primary full dislocation mechanism. (b) Strain map and deformation structure show the formation of C-SB in [112]-oriented pillar. (c) The system has one activate slip system on ACD plane with primary full dislocation mechanism in [123]-oriented pillar. (d) Strain map and deformation structure show formation of C-SB.

- Adding a “Supplementary Note 4 | Dislocation nucleation stress from Frank-Read source, surface step, and surface” (page 26 of SI)

To determine whether dislocation nucleation originates from the sample surface or within the pillar interior, we performed atomistic modeling to compare the dislocation nucleation stresses across three scenarios: free surface, surface step, and Frank-Read source.

We created three perfect pillars with diameters of 22 nm, 32 nm, and 42 nm, each with an aspect ratio of 2.5, to study dislocation nucleation stress at the free surface. These pillars

contain approximately 2 million, 6 million, and 14 million atoms, respectively. To calculate ...

References

1. Maaß, R., Van Petegem, S., Grolimund, D., Van Swygenhoven, H., Kiener, D. & Dehm, G. Crystal rotation in Cu single crystal micropillars: In situ Laue and electron backscatter diffraction. *Appl Phys Lett* **92**, 2006–2009 (2008).
2. Zepeda-Ruiz, L. A., Stukowski, A., Ooppelstrup, T., Bertin, N., Barton, N. R., Freitas, R. & Bulatov, V. V. Atomistic insights into metal hardening. *Nat Mater* 1–6 (2020).
3. Parthasarathy, T. A., Rao, S. I., Dimiduk, D. M., Uchic, M. D. & Trinkle, D. R. Contribution to size effect of yield strength from the stochastics of dislocation source lengths in finite samples. *Scr Mater* **56**, 313–316 (2007).
4. Greer, J. R. & De Hosson, J. Th. M. M. Plasticity in small-sized metallic systems: Intrinsic versus extrinsic size effect. *Prog Mater Sci* **56**, 654–724 (2011).

A point-by-point response to the reviewers' comments

Reviewer #1:

This manuscript presents a systematic investigation that advances our fundamental understanding of slip band formation mechanisms in crystalline materials. Through micropillar compression experiments, the authors have successfully elucidated the distinct formation pathways of confined versus extended slip bands. More importantly, the work introduces groundbreaking defect-mediated deformation mechanisms that operate under varying crystallographic orientations. The revised version significantly strengthens these claims through comprehensive experimental validation and mechanistic analysis, establishing a reliable framework for understanding microstructural evolution. These findings not only provide transformative insights into the atomic-scale processes governing plastic deformation but also establish a new paradigm for controlling strain localization in metallic systems. The quality and impact of this work meet the high standards expected for publication in Nature Communications.

Reply: We sincerely thank the reviewer for the thoughtful review and constructive feedback on our work.

Reviewer #2:

I thank the authors for addressing my concerns. The paper presents a consistent study explaining the formation of different slip bands (confined and extended) in CrCoNi micropillars, supported by very high quality experimental and numerical observations. I enjoyed reading the revised manuscript and the supplementary information provided. Good work!!

Reply: We appreciate the reviewer's valuable suggestions, which have helped us improve the manuscript.